# Epidemiological Study of Lumbar Spinal Stenosis Symptoms: 10-Year Follow-Up in the Community

**DOI:** 10.3390/jcm11195911

**Published:** 2022-10-07

**Authors:** Takahiro Igari, Koji Otani, Miho Sekiguchi, Shin-ichi Konno

**Affiliations:** Department of Orthopaedic Surgery, Fukushima Medical University School of Medicine, Fukushima 960-1295, Japan

**Keywords:** lumbar spinal stenosis, natural history, observational study, epidemiology, quality of life, long follow-up study, questionnaire, Roland-Morris Disability Questionnaire, lumbar spine, local resident, local community

## Abstract

Background: Lumbar spinal stenosis (LSS) is one of the important health problems in an aging society because it can significantly impair quality of life (QOL) and active daily living (ADL). However, the natural history or long-term change of LSS symptoms is still unclear. The purpose of this study was to clarify the 10-year course of lumbar spinal stenosis (LSS) symptoms in community-dwelling residents of more than 1000 people with prospective data collection. Methods: A total of 1149 subjects were analyzed for the time course of LSS symptoms for ten years. LSS symptoms were assessed using a questionnaire specially designed and validated to detect LSS symptoms without image information such as magnetic resonance imaging. Results: The prevalence of positive LSS symptoms was about 16% in the initial survey and 10-year follow-up. Of the subjects who were LSS positive at the initial survey, 40% showed positive LSS symptoms at follow-up and 60% switched to negative LSS symptoms. According to the multivariable logistic regression analysis, severe depression and positive LSS symptoms were extracted as predictors of the presence of LSS symptoms after a 10-year follow-up. Conclusion: The statistical predictor of the presence of LSS symptoms at 10 years was the presence of LSS symptoms at the initial survey; however, 60% of those who were positive for LSS symptoms at the initial survey were not determined to have LSS symptoms at the 10-year follow-up. This was the same result as at the 1-year and 6-year follow-up.

## 1. Introduction

Lumbar spinal stenosis (LSS) is defined as a syndrome of narrowing of the spinal canal, lateral recess, or neural foramina, which are nervous system pathways, and it causes specific symptoms of the lumbar region and lower limbs [1,2,3]. LSS is one of the most serious problems in the elderly because of its high prevalence and negative impact on quality of life (QOL) [3,4]. In addition, it is well known that radiographic findings do not always correlate with symptoms, because anatomic spinal stenosis occurs commonly on imaging in the elderly [5,6]. Therefore, clinical LSS should be diagnosed through subjective symptoms first, and then finally confirmed through objective physical findings supported by radiographic evidence. Furthermore, there are discrepancies between clinical symptoms and imaging findings such as stenotic condition on magnetic resonance image (MRI) in cases of LSS [7,8]. Due to these complexities of the diagnosis of LSS, it has been difficult to conduct a large-scale epidemiologic study of LSS. The prevalence of symptomatic LSS in community-dwelling people was reported using a self-administered, self-reported history questionnaire for LSS (LSS-SSHQ) [2,3,9,10]. This tool was specially designed to detect LSS symptoms without image information such as magnetic resonance imaging (MRI) and has been analyzed in derivation and validation studies and has been confirmed to have acceptable sensitivity, specificity, and reproducibility [11]. Thus, it is used for the identification of LSS based on self-reported patient information alone and considered useful for epidemiological studies to identify the subjects with LSS symptoms.

No study has investigated the long-term (10 years) course of LSS symptoms with more than 1000 subjects in the community. The aim of this study was to clarify the time course of LSS symptoms and their quality of life (QOL) over 10 years in community-dwelling people.

## 2. Materials and Methods

### 2.1. Ethical Approval

This study was approved by the Ethics Committee of our institute (No.1880).

### 2.2. Participants

In this study, the LSS-symptoms survey was combined with annual health checkups of residents enrolled in the National Health Insurance system in Tadami town, Ina village, and Tateiwa village in Fukushima prefecture, Japan. Of the 3367 people who participated in the health checkups, 1862 (697 males, 1165 females; age range, 19–93 years) were followed up for the LSS survey, which corresponds to 21.5% [2] of the 8660 people in the survey area, and 55.0% of those who participated in the health checkups. Ten-year follow-up survey was performed in 2014. In this 10-year follow-up study, a questionnaire was mailed to the subjects for them to complete. Then, a volunteer visited each subject to collect the questionnaires. Participants were excluded if they were unable to walk independently, fill out the questionnaires due to visual impairment, had ever undergone brain or spinal surgery, or had experienced a fracture of the lower extremities in the year previous to the start or the follow-up of the study period [9]. The 1623 people who were younger than 79 years and completely filled LSS-SSHQ in the initial survey were surveyed in this study. Of the 1623 people, 320 did not reply to the questionnaire, 154 had died or moved away. In total, 41 subjects who underwent lumbar spine surgery for 10-year follow-up periods were included as LSS-positive at follow-up. Finally, 1149 people (406 males, 743 females; age range, 30–89 years at 10-year follow-up) replied to the questionnaire, and their answers were analyzed (Figure 1). In this study, the participants voluntarily participated in the LSS survey, and the analysis was also conducted in a state in which there were no omissions in the questionnaire. The percentage of omissions was 75/1862 (4.0%) at the time of the initial survey and 320/1623 (19.7%) at the follow-up survey, which is considered to be a sufficiently reliable filling rate. Written informed consent was obtained from all subjects. 

### 2.3. Diagnosis of LSS Symptoms

To standardize LSS symptoms, the presence of LSS symptoms was diagnosed by a validated LSS diagnostic support tool, the LSS-SSHQ, consisting of 10 yes/no questions (Appendix A). The sensitivity and specificity of the LSS-SSHQ have been confirmed to be 0.855 and 0.791, respectively, in the derivation data and 0.843 and 0.781, respectively, in the validation data. The area under the receiver operating characteristic curve was 0.782. These results show that this tool is reliable for the diagnosis of symptomatic LSS based on self-reported patient information alone and should be useful for epidemiological studies without image findings such as MRI. In the LSS-SSHQ, those who answered “Yes” to all questions (Qs) 1–4 were considered LSS-positive. Those who answered “Yes” to one or more from Qs 1–4 and “Yes” to two or more from Qs 5–10 were also considered LSS-positive [11]. Similar to previous studies, this study classified participants who underwent LSS surgery after the initial survey as LSS positive, regardless of their LSS-SSHQ assessment [9,10].

### 2.4. QOL Evaluation

Japanese version of the Medical Outcomes Study 36-Item Short-Form Health Survey (SF-36) was used to evaluate health-related quality of life (HR-QOL) [12,13]. The SF-36 consists of 8 domains (physical functioning: PF, role-physical: RP, bodily pain: BP, general health: GH, vitality: VT, social functioning: SF, role-emotional: RE, mental health: MH). A Japanese version of the SF-36 is available for both original and norm-based scores [14].

The Roland-Morris Disability Questionnaire (RDQ) is a reliable, validated scale used to measure disability caused by low back pain (LBP) [15,16]. The 24 items in the RDQ can be answered with either “yes” or “no”. A Japanese version of the RDQ is available for both original and norm-based scores. The norm-based score was calculated based on the average score of Japanese people with LBP.

In the norm-based score of both eight domains of SF-36 and RDQ, the average score and one-standard deviation were 50 and 10, respectively, and a score of more than 50 points meant better QOL compared to the national norm [14,17]. In this study, the norm-based scores of the SF-36 were used to rule out 10-year effects.

### 2.5. Predictive Factors of the Presence of LSS Symptoms at 10-Year Follow-Up

In the initial survey, we investigated clinical history of participants about hypertension, cardiovascular disease, cerebrovascular disease, respiratory disease, diabetes mellitus and smoking status by six public health nurses [2]. All participants were also checked for their smoking status, and their pack-years (Brinkman index) were calculated. The depressive symptoms were assessed using the MH scores in SF-36 [18]. A MH score of <52 was defined severe, 52–59 as moderate, 60–67 as mild, and >67 as none. The OA of the knee and hip was evaluated by physicians through Altman’s criteria [19,20].

### 2.6. Statistical Analysis

Subjects were divided into two groups such as the positive LSS symptoms (LSS (+)) and the negative LSS symptoms (LSS (−)) at the initial survey and 10-year follow-up. From the point of view of time course of LSS symptoms, the subjects were finally divided into 4 groups by the change from initial to the follow-up period; (+) → (+), (+) → (−), (−) → (+) and (−) → (−). Statistical testing was done using IBM SPSS Statistics (ver. 23, SPSS Inc., Chicago, IL, USA). Paired t-test was used to compare the change of RDQ and SF-36 norm-based scores between the initial survey and the 10-year follow-up in each four groups except for operation cases. Furthermore, the predictive factors for the presence of LSS symptoms at 10-year follow-up was examined using the multivariable logistic regression analysis. Since LSS-SSHQ is judged by lower limb symptoms, the effect of hip or knee OA might be ruled out. Therefore, model 1 analyzed all subjects regardless of the presence or absence of hip or knee OA, and model 2 analyzed subjects excluding hip or knee OA at the initial survey. Values of *p* < 0.05 were considered statistically significant.

## 3. Results

### 3.1. Time Course of LSS Symptoms

The demographic data in the initial survey are shown in Table 1. In total, 187 out of 1149 participants (16.3%) had positive LSS symptoms and they increased with age in the initial survey. The prevalence of positive LSS symptoms was the same between the initial survey and 10-year follow-up. Of the 187 subjects who were LSS positive at the initial survey, 71 (38.0%) showed positive LSS symptoms at follow-up and 116 (62.0%) switched to negative LSS symptoms. There were 962 subjects who showed negative LSS symptoms at the initial survey. Of the 962 subjects, 119 (12.4%) switched to positive LSS symptoms, while 843 (87.6%) remained in negative LSS symptoms at the 10-year follow-up period (Figure 2). 

To compare with age, the prevalence of positive LSS symptoms at the initial survey that switched to negative LSS symptoms at follow-up decreased with age. On the other hand, the prevalence of negative LSS symptoms at the initial survey that switched to positive LSS symptoms at follow-up increased with age (Table 2).

### 3.2. Relationship between LBP-Related QOL and the Time Course of LSS Symptoms

The changes of the RDQ score according to the time curse of LSS symptoms were shown in Table 3. At the initial survey, the RDQ sores were lower in the participant with positive LSS than those with negative LSS symptoms. In the participants with LSS symptoms at the initial survey, (+) → (+) and (+) → (−), there was no difference of the RDQ score between two groups at the initial survey. At the 10-year follow-up periods, the RDQ sores were lower in the participant with positive LSS than those with negative LSS symptoms. In the participants with LSS symptoms at the 10-year follow-up period; (+) → (+) and (−) → (+), there was no difference of the RDQ score between two groups at follow-up periods. In the participant with negative LSS symptom at follow-up period, the RDQ score improved compared with those at the initial survey. These results suggested that the duration of LSS symptoms might not influence the RDQ score but the daily disability occurred due to the presence of LSS symptoms. 

### 3.3. Relationship between HR-QOL and the Time Course of LSS

All eight domains of SF-36 scores at follow-up were worse compared to those scores at the initial survey regardless of the time course of LSS symptoms. Although most of the score reduction was statistically apparent, the decrease in scores in the group with negative LSS symptoms at follow-up appears to be smaller than the decrease in scores in the group with positive LSS symptoms at follow-up (Table 4).

### 3.4. Predictive Factor of the Presence of LSS Symptoms during the 10-Year Follow-Up Period

RDQ norm-based score less than the norm, knee OA, severe depressive symptom and positive LSS symptoms in model 1 and pack-years (Brinkman index), severe depressive symptom and positive LSS symptoms in model 2 were detected as predictive factors of the presence of LSS symptoms after 10 years (Table 5). Severe depression and positive LSS symptoms were extracted as predictors of the presence of LSS symptoms after 10-year follow-up in both models.

## 4. Discussion

We started an epidemiological study of LSS from 2004 [2]. In the initial survey, the prevalence of LSS symptoms increased with age, and the presence of LSS symptoms reduced low back pain-related QOL and health-related QOL. Up to now, there is few studies regarding time course of LSS in the community. This study was conducted to reveal the time course of LSS symptoms in community-dwelling people over a 10-year period. 

It has been reported that the clinical symptoms of LSS did not develop in more than 60% of patients who received conservative treatment for 10 years in a clinical setting [21]. Similarly, in the present study, 62.0% of subjects who showed positive LSS symptoms at the initial survey and negative at 10-year follow-up. Overall, about 60% of people recovered from LSS symptoms after 10 years, and younger people were particularly likely to recover from LSS symptoms. On the other hand, before this study was initiated, it was expected that the frequency of LSS symptoms switching from positive to negative at the 10-year follow-up would decrease in the same cohort compared to the 1-year and 6-year follow-ups. In fact, however, the frequency of LSS symptoms switching from positive to negative at the 10-year follow-up was similar at the 1-year and 6-year follow-ups. These results support the description of North American Spine Society Evidence-Based Clinical Guidelines which “the natural history of patients with clinically mild to moderately symptomatic degenerative lumbar stenosis can be favorable in about one-third to one-half of patients” [22], even with a 10-year follow-up period. 

QOL of the elderly is declining year by year [14,17]. So, in the current study, keeping in mind that the absolute value of QOL decreases year by year, we investigated the relationship between changes in LSS over 10 years and QOL based on norm-based scores by gender and age group. Changes in LBP-related QOL (RDQ norm-based score) for changes in LSS symptoms over a 10-year period and changes in HR-QOL (eight domains of SF-36) for changes in LSS symptoms over a 10-year period were different. RDQ norm-based score improved when the LSS symptoms disappeared and worsened when the LSS symptoms appeared. This result indicates that RDQ score improved with the relief of LSS symptoms and tended to worsen with the development of LSS symptoms. On the other hand, almost all eight domains of SF-36 scores at follow-up were statistically worse compared to those scores at the initial survey regardless of the time course of LSS symptoms. These reductions tended to be relatively mild in the group without LSS symptoms at follow-up, especially in the group without LSS symptoms in the initial survey. This result indicates that the presence of LSS symptoms not only directly affects HR-QOL, but also the presence of past LSS symptoms affects HR-QOL even if there is no current LSS symptoms. In other words, LSS symptoms have a strong influence on HR-QOL [2,9,10]. 

The correlated factors of LSS were reported hypertension, diabetes mellitus, osteoarthritis and depressive symptoms [3,23]. In this study, according to analyses of all subjects and the subjects without either knee or hip OA, severe depressive symptom and the presence of LSS symptoms in the initial survey were detected the predictive factors of the presence of LSS symptoms over 10 years and there were no significant differences in the comorbidities. 

The odds rate of a low RDQ score as LSS symptoms positive after 10 years was similarly to one- and six-year follow-up study [9,10]. Although Knee OA may be a predictive factor, the results of the LSS-SSHQ may have included symptoms of knee OA. These results mean that a low RDQ score may predict the persons with LSS symptoms one year and ten years later. 

This study had several limitations. First, it was a problem of judgment of LSS-symptoms. In this study, LSS-SSHQ was used to standardize subjective LSS-symptoms. However, physical findings, including neurological and imaging findings such as MRI findings, were not taken into account in determining LSS-symptoms, and LSS-like symptoms caused by diseases other than LSS could not be completely excluded. Second, this research was conducted in a rural area. Third, participants in this study were volunteers. The research area and the use of volunteers might have introduced selection bias. Fourth, the severity of LSS, for example pain and paralysis, was not analyzed. Fifth, the history of treatment for LSS was not evaluated. Nonetheless, this study might still have value because it is the largest study (more than 1000 people) with such a long-term follow-up (10-year) of LSS in the community. Additionally, the results of this study demonstrate that the description of North American Spine Society Evidence-Based Clinical Guidelines which “the natural history of patients with clinically mild to moderately symptomatic degenerative lumbar stenosis can be favorable in about one-third to one-half of patients” [22] is valid at 10-year follow-up. In addition, further studies are needed to confirm the risk factors for LSS to maintain the health of an aging society.

## 5. Conclusions

Overall, in 62.0% of the positive LSS symptom group at the initial survey, symptoms improved over 10 years, while 12.4% of the negative LSS symptoms group showed LSS symptoms. LBP-related QOL and LSS symptoms were correlated over the 10-year period. However, HR-QoL did not show such a clear relationship with LSS symptoms. The presence of LSS symptoms and severe depression may predict the presence of LSS symptoms after ten years.

## Figures and Tables

**Figure 1 jcm-11-05911-f001:**
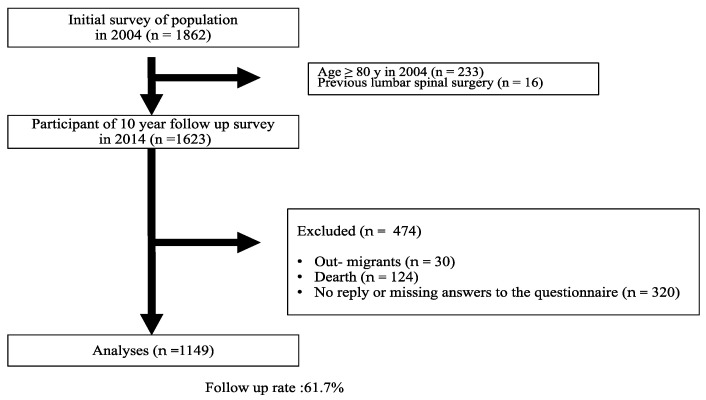
Registration protocol of the questionnaire.

**Figure 2 jcm-11-05911-f002:**
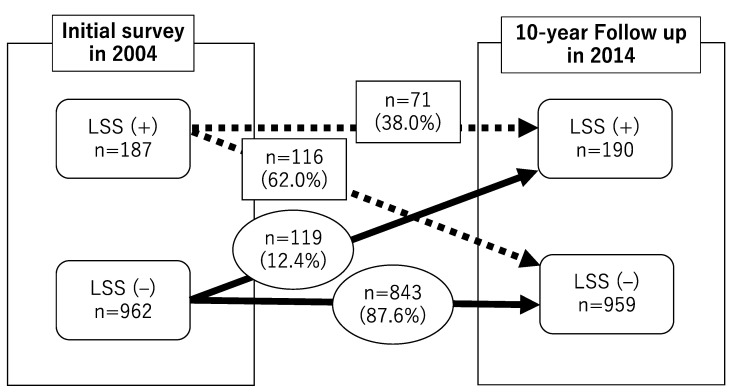
The number of LSS (+) and (−) cases in 2004 and 2014 and the time course of LSS. LSS, lumbar spinal stenosis.

**Table 1 jcm-11-05911-t001:** Demographic characteristics of participants at initial survey.

	Total (n = 1149)	Number of Missing Data
**Age (y) (mean ± SD)**	63.0 ± 11.6	0
**Age in 2004 (n [%])**		0
≤49	155 (13.5)	
50–59	202 (17.6)	
60–69	414 (36.0)	
70–79	378 (32.9)	
**Sex (n [%])**		0
Male	406 (35.3)	
Female	743 (64.7)	
**BMI (kg/m^2^) (n [%])**		79
<18.5	32 (30.0)	
≥18.5, <25	692 (64.7)	
≥25, <30	313 (29.3)	
≥30	33 (3.1)	
**Prevalence of LSS (n [%])**	189 (16.4)	0
**Smoking (n [%])**	205 (19.2)	79
**Comorbidities (n [%])**		79
Respiratory	3 (0.3)	
Diabetes mellitus	54 (5.0)	
Cardiovascular	83 (7.8)	
Cerebrovascular	8 (0.7)	
Hypertension	358 (33.5)	
**Depressive symptoms (MHI-5) (n [%])**		85
None	726 (68.2)	
Mild	121 (11.4)	
Moderate	96 (9.0)	
Severe	122 (11.5)	
**RDQ standardized score (n [%])**		22
≥50	890 (79.0)	
<50	237 (21.0)	
**Prevalence of Knee OA (n [%])**	303 (28.4)	83
**Prevalence of Hip OA (n [%])**	59 (5.2)	5

Abbreviations: SD, standard deviation; BMI, body mass index; LSS, lumbar spinal stenosis; NHI-5, Mental Health Inventory-5, RDQ, Roland-Morris Disability Questionnaire; OA, osteoarthritis.

**Table 2 jcm-11-05911-t002:** Time course of LSS and age.

	Time Course of LSS Symptoms from 2004 to 2014				
Age in 2004 (years)	(+) → (+) n = 71	(+) → (−) n = 116	(−) → (+) n = 119	(−) → (−) n = 843	n
≦ 49	0 (0)	4 (2.6)	9 (5.8)	142 (91.6)	155
50–59	7 (3.5)	15 (7.4)	17 (8.4)	163 (80.7)	202
60–69	26 (6.3)	45 (10.9)	48 (11.6)	295 (71.3)	414
70–79	38 (10.1)	52 (13.8)	45 (11.9)	243 (64.3)	378

Abbreviations: LSS, lumbar spinal stenosis.

**Table 3 jcm-11-05911-t003:** Time course of LSS and RDQ norm-based scores.

Time Course of LSS	RDQ Norm-Based Score		
	Initial Survey	10-Year Follow-Up	*p* Value
(+) → (+)	48.2 ± 8.4	51.4 ± 9.8	0.067
(+) → (−)	49.5 ± 9.5	55.9 ± 9.1	<0.0001
(−) → (+)	53.0 ± 9.0	51.1 ± 10.6	0.3897
(−) → (−)	56.1 ± 6.3	58.4 ± 5.1	<0.0001
(+) → (+)	48.2 ± 8.4	51.4 ± 9.8	0.067

Abbreviations: LSS, lumbar spinal stenosis; RDQ, Roland-Morris Disability Questionnaire.

**Table 4 jcm-11-05911-t004:** Time course of LSS symptoms and eight domains of SF-36 norm-based score.

	Time Course of LSS Symptoms	Norm-Based Score		*p* Value
		Initial Survey	10-Year Follow-Up	
PF	(+) → (+)	47.9 ± 13.8	24.7 ± 22.9	0.0002
	(+) → (−)	48.4 ± 10.3	36.9 ± 19.4	0.0017
	(−) → (+)	47.9 ± 15.6	31.7 ± 17.2	0.0013
	(−) → (−)	50.8 ± 11.7	45.1 ± 14.7	<0.0001
RP	(+) → (+)	43.4 ± 11.9	26.9 ± 15.2	0.0008
	(+) → (−)	46.9 ± 11.8	37.2 ± 18.1	0.0122
	(−) → (+)	44.8 ± 13.1	39.1 ± 17.1	0.0484
	(−) → (−)	50.0 ± 10.0	45.1 ± 13.2	<0.0001
BP	(+) → (+)	47.7 ± 8.7	38.0 ± 8.8	0.0008
	(+) → (−)	43.2 ± 7.6	42.1 ± 9.2	0.5483
	(−) → (+)	48.4 ± 10.1	35.9 ± 7.7	<0.0001
	(−) → (−)	50.5 ± 10.1	46.8 ± 10.8	<0.0001
GH	(+) → (+)	45.7 ± 11.6	44.4 ± 7.2	0.5559
	(+) → (−)	44.8 ± 8.3	47.6 ± 12.8	0.0243
	(−) → (+)	48.6 ± 9.6	44.2 ± 6.7	0.1717
	(−) → (−)	51.3 ± 9.2	51.3 ± 9.5	0.2334
VT	(+) → (+)	51.7 ± 6.8	44.2 ± 7.0	0.0092
	(+) → (−)	49.5 ± 9.7	48.2 ± 11.5	0.4245
	(−) → (+)	49.4 ± 11.5	45.3 ± 8.7	0.0884
	(−) → (−)	53.1 ± 8.7	50.7 ± 9.5	0.0094
SF	(+) → (+)	49.1 ± 10.0	39.0 ± 13.7	<0.0001
	(+) → (−)	47.7 ± 11.7	46.0 ± 15.0	0.5749
	(−) → (+)	48.1 ± 11.1	41.7 ± 11.4	0.0446
	(−) → (−)	50.8 ± 9.3	48.8 ± 11.2	0.0049
RE	(+) → (+)	49.9 ± 10.7	32.7 ± 15.3	0.0009
	(+) → (−)	46.9 ± 12.4	42.5 ± 17.1	0.2191
	(−) → (+)	45.2 ± 13.9	35.5 ± 11.3	0.0123
	(−) → (−)	50.7 ± 9.6	46.7 ± 12.7	<0.0001
MH	(+) → (+)	46.7 ± 12.9	45.1 ± 7.9	0.6228
	(+) → (−)	45.7 ± 9.6	47.2 ± 10.9	0.4428
	(−) → (+)	46.6 ± 9.8	43.8 ± 9.3	0.0889
	(−) → (−)	50.4 ± 9.2	51.3 ± 9.9	0.0841

Abbreviations: PF, physical functioning; RP, role-physical; BP, bodily pain; GH, general health; perception; VT, vitality; SF, social functioning; RE, role-emotional; MH, mental health.

**Table 5 jcm-11-05911-t005:** The predictive factors of the occurrence of LSS symptoms during the 10-year follow-up period.

		Model 1 (n = 1149)			Model 2 (n = 730)		
Category 1	Category 2	OR	95%CI	*p* Value	OR	95%CI	*p* Value
Gender	Female	0.841	0.546–1.295	0.4315	1.07	0.627–1.826	0.8701
Age (years)	≤49	0.623	0.257–1.511	0.295	0.615	0.230–1.644	0.3322
	50–59	reference			reference		
	60–69	1.437	0.803–2.751	0.2221	1.672	0.809–3.455	0.1654
	70–79	1.696	0.943–3.051	0.0777	1.673	0.776–3.607	0.1891
BMI	<18.5	reference			reference		
	18.5–24.9	4.565	0.993–20.975	0.0551	3.2 × 10^8^	-	0.9981
	25.0–29.9	3.929	0.838–18.428	0.0826	3.4 × 10^8^	-	0.9981
	30.0≤	3.044	0.426–2.764	0.2673	1.2 × 10^8^	-	0.9982
RDQ norm-based score	50≤	reference			reference		
	<50	1.892	1.230–2.910	0.0037	1.405	0.699–2.824	0.3404
Knee OA	Positive	1.984	1.302–2.964	0.013			
Hip OA	Positive	1.480	0.699–3.135	0.3056			
Comorbidities	Respiratory	1.706	0.093–31.452	0.7193	2.7 × 10^−9^	-	0.9993
	Diabetes Mellitus	1.345	0.627–2.886	0.4464	0.839	0.232–3.033	0.7883
	Cardiovascular	0.589	0.295–1.176	0.1335	0.581	0.159–2.216	0.4116
	Cerebrovascular	0.661	0.0072–6.088	0.7151	2.7 × 10^−9^	-	0.9993
	Hypertension						
Smoking	Pack-years	1.000	1.000–1.001	0.2164	1.001	1.000–1.001	0.0187
Depressive symptoms	None	reference			reference		
	Mild	1.534	0.885–2.661	0.1372	1.531	0.719–3.260	0.2688
	Moderate	0.809	0.389–1.681	0.570	0.949	0.346–2.603	0.919
	Severe	2.379	1.415–3.999	0.0011	3.487	1.709–7.113	0.0006
LSS symptoms in the initial survey	Positive	2.749	1.765–4.282	<0.0001	3.732	1.758–7.802	0.0005

Abbreviations: OR, odds ratio; CI, confidence interval; BMI, body mass index; RDQ, Roland-Morris Disability Questionnaire; OA, osteoarthritis; LSS, lumbar spinal stenosis.

## Data Availability

The data presented in this study could be available on request from the corresponding author. The data are not publicly available because the underlying data were obtained from the collaboration with the local government and contain sensitive information on individuals including gender, age and self-reported data, and sharing these data openly is prohibited by the local government and Fukushima Medical University Ethics Committee.

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
