# Peer review of "Epidemiological Study of Lumbar Spinal Stenosis Symptoms: 10-Year Follow-Up in the Community"

_jcm, 2022, doi:10.3390/jcm11195911_

Round 1

Reviewer 1 Report

The purpose of this study was to clarify the 10-year course of lumbar spinal stenosis (LSS) symptoms in community-dwelling residents of more than 1,000 people with prospective data collection.

Authors have undertaken great work, although it is commonly known, that lumbar spinal stenosis has a negative impact on quality of life compared with other comorbidities both in Japan and in other well-developed countries, according to the descriptions in a vast majority of papers cited in Refs. The question arises, what is a novum in a manuscript under consideration?

They state, that they used a questionnaire specially designed and validated to detect LSS symptoms (Roland-Morris Disability Questionnaire) without image information such as magnetic resonance imaging. Using this tool, it is very difficult to confirm definitely the symptoms of LSS, no imaging or clinical studies including instrumental have been used, giving the opportunity to assume other possible back pain disorders influencing the results of the questionnaire.

Using this tool, it is very difficult to confirm definitely the symptoms of LSS, no imaging or clinical studies including instrumental have been used, giving the opportunity to assume other possible back pain disorders influencing the results of the questionnaire.

Obtained results are very easy for predicting, in subjects who were positive LSS at the initial survey, 40% showed positive LSS symptoms at follow-up and 60% switched to negative  LSS symptoms. Similar results were also described in the literature presented in Refs. and do not bring any news to medical knowledge.

The low-quality study design expresses the most in the study limitations in the end of the Discussion section. For example, the authors listed honestly, that participants in this study were volunteers from the ryral ares, the research area and the use of volunteers might have introduced selection bias.

The low-quality study design expresses itself the most in the study limitations at the end of the Discussion section. For example, the authors listed honestly, that participants in this study were volunteers from rural areas, the research area and the use of volunteers might have introduced selection bias.

Author Response

Reviewer 1

Thank you for your important comments, which have enabled us to improve the manuscript. We have provided our point-by-point responses below.

  1. They state, that they used a questionnaire specially designed and validated to detect LSS symptoms (Roland-Morris Disability Questionnaire) without image information such as magnetic resonance imaging. Using this tool, it is very difficult to confirm definitely the symptoms of LSS, no imaging or clinical studies including instrumental have been used, giving the opportunity to assume other possible back pain disorders influencing the results of the questionnaire. Using this tool, it is very difficult to confirm definitely the symptoms of LSS, no imaging or clinical studies including instrumental have been used, giving the opportunity to assume other possible back pain disorders influencing the results of the questionnaire.

Response: Thank you very much for your critical comments about the questionnaire (LSS-SSHQ), which we used in this study to assess LSS symptoms (but not to diagnose LSS). Patient-based assessment methods such as patient-reported questionnaires and screening tools for disease have been developed and validated in previous studies, and enable evaluation of individual conditions without neurological or imaging findings (ref. 11). Accordingly, the LSS-SSHQ is a simple and useful method for selecting a suitable set of subjects for multicenter and epidemiological studies. As the LSS-SSHQ used in this study had been validated previously in comparison with neurological and imaging findings, we considered that it was suitable for use in health checkups.

  1. Obtained results are very easy for predicting, in subjects who were positive LSS at the initial survey, 40% showed positive LSS symptoms at follow-up and 60% switched to negative LSS symptoms. Similar results were also described in the literature presented in Refs. and do not bring any news to medical knowledge.

Response: As you mentioned, our results are similar to those reported in the clinically based study of Minamide (ref. 21). However, that study and most previous studies were hospital-based, and therefore the results are unlikely to be generalizable. For this reason, population-based studies are required.

It is a strength of this study that progression of LSS symptoms was evaluated for 10 years in a longitudinal study of more than 1000 community-dwelling adults, to demonstrate the likely time course of LSS symptoms in the real world. The occurrence of LSS increases with age; however, according to the present results, LSS symptoms did not persist throughout the 10 year follow-up period in all patients. It is necessary to conduct further studies to evaluate predictors of LSS and how LSS can be prevented.

Another strength of our research is that the same results were found at 1-year (ref. 9), 6-year (ref. 10), and 10-year follow-ups. We were surprised that the results were the same at 1, 6, and 10 years; we had expected to find fewer patients with LSS symptoms switching from positive to negative and more patients switching from negative to positive at 10 years, compared with the 1-year and 6-year follow-ups. We think that these two points are worth reporting.

  1. The low-quality study design expresses the most in the study limitations in the end of the Discussion section. For example, the authors listed honestly, that participants in this study were volunteers from the rural areas, the research area and the use of volunteers might have introduced selection bias.

Response: Thank you for mentioning these points. In this study, the LSS-symptoms survey was combined with annual health checkups of residents enrolled in the National Health Insurance system in Tadami town, Ina village, and Tateiwa village in Fukushima prefecture, Japan. Of the 3367 people who participated in the health checkup, 1862 were followed up for the LSS survey, which corresponds to 21.5% (ref. 2) of the 8660 people in the survey area, and 55.0% of those who participated in the health checkup. Conducting the surveys together with the health checkups enabled us to link the survey results with various data related to health checkups, such as comorbidities. We have added the following sentences in response to this comment.

A total of 1862 people (697 males, 1165 females; age range, 19–93 years) agreed to participate and were interviewed in 2004 [2,9,10]. All were local inhabitants of Tadami town, Ina village, and Tateiwa village in Fukushima prefecture, Japan.

In this study, the LSS-symptoms survey was combined with annual health checkups of residents enrolled in the National Health Insurance system in Tadami town, Ina village, and Tateiwa village in Fukushima prefecture, Japan. Of the 3367 people who participated in the health checkups, 1862 (697 males, 1165 females; age range, 19–93 years) were followed up for the LSS survey, which corresponds to 21.5% [2] of the 8660 people in the survey area, and 55.0% of those who participated in the health checkups.

Reviewer 2 Report

In this paper entitled "Epidemiological Study Pf Lumbar Spinal Stenosis symptoms: 10-year Follow-up in the Community" Igari et al. have collected information on patients with spinal stenosis syndrome over a course of 10 years.

They concluded that the statistical predictor of the presence of LSS symptoms at 10 years was the presence of LSS symptoms at the initial survey, however, 60% of those who were positive for LSS symptoms at the initial survey were not determined to have LSS symptoms at the 10-year follow-up.

This is well written paper, with minimal grammar issues, with a robust scientific structure and adequate conclusions. This is also a very common topic with potential repercussions on clinical practice.

Author Response

Reviewer 2

Thank you very much for your kind comments, which have encouraged us to continue our research into lumbar spinal stenosis, which is common in the elderly and greatly affects their QOL.

Round 2

Reviewer 1 Report

 A comparison of the old and new versions of the manuscript leads to the conclusions about the minor changes introduced by the Authors.

The authors answered as it was possible to my questions and remarks.

They developed the description of the study group of patients and the location of the participants they belong to, methods are still questionaries, and the study design didn’t change greatly.  They explain, that patient-reported questionnaires and screening tools for the disease have been developed and validated in previous studies, and enable the evaluation of individual conditions without neurological or imaging findings (ref. 11). As I mentioned before, using this tool, is very difficult to confirm definitely the symptoms of LSS, no imaging or clinical studies including instrumental have been used (or reported on their using), giving the opportunity to assume other possible back pain disorders influencing the results of the questionnaire. I think that the Authors should honestly express more of the above opinion in the study limitations.

I noticed, that obtained results are very easy for predicting, in subjects who were positive LSS at the initial survey, 40% showed positive LSS symptoms at follow-up and 60% switched to negative LSS symptoms. Similar results were also described in the literature presented in Refs. and did not bring any news to the medical knowledge. The authors admitted, that results are similar to those reported in the clinically based study of Minamide (ref. 21). However, that study and most previous studies were hospital-based, and therefore the results are unlikely to be generalizable. For this reason, population-based studies are required.            I agree with their statement.

However, they conclude with two new strengths of their research, that is the progression of LSS symptoms was evaluated for 10 years in a longitudinal study of more than 1000 community-dwelling adults, to demonstrate the likely time course of LSS symptoms in the real world. The occurrence of LSS increases with age; however, according to the present results, LSS symptoms did not persist throughout the 10-year follow-up period in all patients. It is necessary to conduct further studies to evaluate predictors of LSS and how LSS can be prevented.         I agree.

Another strength of their research is that the same results were found at 1-year (ref. 9), 6-year (ref. 10), and 10-year follow-ups. They were surprised that the results were the same at 1, 6, and 10 years; they had expected to find fewer patients with LSS symptoms switching from positive to negative and more patients switching from negative to positive at 10 years, compared with the 1-year and 6-year follow-ups. They think that these two points are worth reporting.

I absolutely agree, I advise building the additional short part of the Discussion on the above issues and perhaps include a short sentence in the Abstract.

Their additional explanation in the text: In this study, the LSS-symptoms survey was combined with annual health checkups of residents enrolled in the National Health Insurance system in Tadami town, Ina village, and Tateiwa village in Fukushima prefecture, Japan. Of the 3367 people who participated in the health checkups, 1862 (697 males, 1165 females; age range, 19–93 years) were followed up for the LSS survey, which corresponds to 21.5% [2] of the 8660 people in the survey area, and 55.0% of those who participated in the health checkups.     …partially defends the study design and the whole hard work based on the questionnaire results. If the Authors are convinced that participants from the rural areas understood the questions well and responded to them honestly, I suggest putting such information in the M&M section.

I think that this paper will qualify better to be published in JCM considering the further minor corrections although its novelty is still average.

I leave the final decision to the JCM Academic Editor.

Author Response

Reviewer 1

Thank you for your kind comments, which have enabled us to improve the manuscript. We have provided our point-by-point responses below, again.

  1. They developed the description of the study group of patients and the location of the participants they belong to, methods are still questionaries, and the study design didn’t change greatly. They explain, that patient-reported questionnaires and screening tools for the disease have been developed and validated in previous studies, and enable the evaluation of individual conditions without neurological or imaging findings (ref. 11). As I mentioned before, using this tool, is very difficult to confirm definitely the symptoms of LSS, no imaging or clinical studies including instrumental have been used (or reported on their using), giving the opportunity to assume other possible back pain disorders influencing the results of the questionnaire. I think that the Authors should honestly express more of the above opinion in the study limitations.

Response: Thank you very much for your suggestion. We entirely agree with your suggestion. In response to this comment, we have added the following sentences to the section of limitation (Line 260).

First, it was a problem of judgment of LSS-symptoms. In this study, LSS-SSHQ was used to standardize subjective LSS-symptoms. However, physical findings, including neurological and imaging findings such as MRI findings, were not taken into account in determining LSS-symptoms, and LSS-like symptoms caused by diseases other than LSS could not be completely excluded.

  1. I noticed, that obtained results are very easy for predicting, in subjects who were positive LSS at the initial survey, 40% showed positive LSS symptoms at follow-up and 60% switched to negative LSS symptoms. Similar results were also described in the literature presented in Refs. and did not bring any news to the medical knowledge. The authors admitted, that results are similar to those reported in the clinically based study of Minamide (ref. 21). However, that study and most previous studies were hospital-based, and therefore the results are unlikely to be generalizable. For this reason, population-based studies are required.

I agree with their statement.

Response: Thank you very much for your understanding.

  1. However, they conclude with two new strengths of their research, that is the progression of LSS symptoms was evaluated for 10 years in a longitudinal study of more than 1000 community dwelling adults, to demonstrate the likely time course of LSS symptoms in the real world. The occurrence of LSS increases with age; however, according to the present results, LSS symptoms did not persist throughout the 10-year follow-up period in all patients. It is necessary to conduct further studies to evaluate predictors of LSS and how LSS can be prevented.

I agree.

Another strength of their research is that the same results were found at 1-year (ref. 9), 6-year (ref. 10), and 10-year follow-ups. They were surprised that the results were the same at 1, 6, and 10 years; they had expected to find fewer patients with LSS symptoms switching from positive to negative and more patients switching from negative to positive at 10 years, compared with the 1-year and 6-year follow-ups. They think that these two points are worth reporting.

I absolutely agree, I advise building the additional short part of the Discussion on the above issues and perhaps include a short sentence in the Abstract.

Response: Thank you very much for your understanding and suggestion. In response to this comment, we have added the following sentences to Abstract and the section of Discussion.

  • Abstract (Line 25); This was the same result at the 1-year and 6-year follow-up.

  • Discussion (Line 222); Overall, about 60% of people recovered from LSS symptoms after 10 years, and younger people were particularly likely to recover from LSS symptoms. These results support the description of North American Spine Society Evidence-Based Clinical Guidelines which “the natural history of patients with clinically mild to moderately symptomatic degenerative lumbar stenosis can be favorable in about one-third to one-half of patients” [22].

Overall, about 60% of people recovered from LSS symptoms after 10 years, and younger people were particularly likely to recover from LSS symptoms. On the other hand, before this study was initiated, it was expected that the frequency of LSS symptoms switching from positive to negative at the 10-year follow-up would decrease in the same cohort compared to the 1-year and 6-year follow-ups. In fact, however, the frequency of LSS symptoms switching from positive to negative at the 10-year follow-up was similar at the 1-year and 6-year follow-ups. These results support the description of North American Spine Society Evidence-Based Clinical Guidelines which “the natural history of patients with clinically mild to moderately symptomatic degenerative lumbar stenosis can be favorable in about one-third to one-half of patients” [22], even with a 10-year follow-up period.

  • Discussion (Line 268); Nonetheless, this study might still have value because it is the largest study with such a long-term follow-up of LSS in the community. In addition, further studies are needed to confirm the risk factors for LSS to maintain the health of an aging society.

Nonetheless, this study might still have value because it is the largest study (more than 1000 people) with such a long-term follow-up (10-year) of LSS in the community. And the results of this study demonstrate that the description of North American Spine Society Evidence-Based Clinical Guidelines which “the natural history of patients with clinically mild to moderately symptomatic degenerative lumbar stenosis can be favorable in about one-third to one-half of patients” [22] is valid at 10-year follow-up. In addition, further studies are needed to confirm the risk factors for LSS to maintain the health of an aging society.

  1. Their additional explanation in the text: In this study, the LSS-symptoms survey was combined with annual health checkups of residents enrolled in the National Health Insurance system in Tadami town, Ina village, and Tateiwa village in Fukushima prefecture, Japan. Of the 3367 people who participated in the health checkups, 1862 (697 males, 1165 females; age range, 19–93 years) were followed up for the LSS survey, which corresponds to 21.5% [2] of the 8660 people in the survey area, and 55.0% of those who participated in the health checkups. …partially defends the study design and the whole hard work based on the questionnaire results. If the Authors are convinced that participants from the rural areas understood the questions well and responded to them honestly, I suggest putting such information in the M&M section.

Response: Thank you very much for your kind comments and suggestion. In response to this suggestion, we have added the following sentences to the section of Materials and Methods.

Materials and Methods (Line 74); Finally, 1149 people (406 males, 743 females; age range, 30-89 years at 10-year follow-up) replied to the questionnaire, and their answers were analyzed (Figure 1). Written in-formed consent was obtained from all subjects.

Finally, 1149 people (406 males, 743 females; age range, 30-89 years at 10-year follow-up) replied to the questionnaire, and their answers were analyzed (Figure 1). In this study, the participants voluntarily participated in the LSS survey, and the analysis was also conducted in a state in which there were no omissions in the questionnaire. The percentage of omissions was 75/1862 (4.0%) at the time of the initial survey and 320/1623 (19.7%) at the follow-up survey, which is considered to be a sufficiently reliable filling rate. Written in-formed consent was obtained from all subjects.
